# Developments of Space Debris Laser Ranging Technology Including the Applications of Picosecond Lasers

**Haifeng Zhang** [1,2], **Mingliang Long** [1,*] , **Huarong Deng** [1], **Shaoyu Cheng** [1], **Zhibo Wu** [1,2], **Zhongping Zhang** [1,2,3,*], **Ali Zhang** [4] and **Jiantao Sun** [4]

1   Shanghai Astronomical Observatory, Chinese Academy of Sciences, Shanghai 200030, China; hfzhang@shao.ac.cn (H.Z.); dhr@shao.ac.cn (H.D.); chengshaoyu@shu.edu.cn (S.C.); wzb@shao.ac.cn (Z.W.)
2   Key Laboratory of Space Object and Debris Observation, Chinese Academy of Sciences, Nanjing 210008, China
3   State Key Laboratory of Precision Spectroscopy, East China Normal University, Shanghai 200062, China
4   Xinjiang Astronomical Observatory, Chinese Academy of Sciences, Urumchi 830011, China; Zhangal@xao.ac.cn (A.Z.); sunjt@xao.ac.cn (J.S.)
*   Correspondence: longmingliang@shao.ac.cn (M.L.); zzp@shao.ac.cn (Z.Z.)

**Abstract:** Debris laser ranging (DLR) is receiving considerable attention as an accurate and effective method of determining and predicting the orbits of space debris. This paper reports some technologies of DLR, such as the high pulse repetition frequency (PRF) laser pulse, large-aperture telescope, telescope array, multi-static stations receiving signals. DLR with a picosecond laser at the Shanghai Astronomical Observatory is also presented. A few hundred laps of space debris laser-ranging measurements have been made. A double-pulse picosecond laser with an average power of 4.2 W, a PRF of 1 kHz, and a wavelength of 532 nm has been implemented successfully in DLR, it's the first time that DLR technology has reached a ranging precision at the sub-decimeter level. In addition, the characteristics of the picosecond-pulse-width laser transmission with the advantages of transmission in laser ranging were analyzed. With a mode of the pulse-burst picosecond laser having high average power, the DLR system has tracked small debris with a radar cross-section (RCS) of 0.91 m$^2$ at a ranging distance up to 1726.8 km, corresponding to an RCS of 0.1 m$^2$ at a distance of 1000 km. These works are expected to provide new technologies to further improve the performance of DLR.

**Keywords:** space debris laser ranging; single-photon detection; picosecond laser; pulse-bursts

## 1. Introduction

Human activities in space have increased with the development of space technology, resulting in many uncontrollable artificial satellites, rocket bodies, and spacecraft in space [1–7]. Space debris, also known as space rubbish, refers to all human-made objects in orbit except for spacecraft in normal operation and includes rocket bodies and satellite bodies that have completed missions, rocket jets, objects discarded during the execution of space missions, and fragments generated by collisions between space objects. According to a report issued by the 32nd Inter-Agency Space Debris Coordination Committee, there are hundreds of millions of pieces of space debris larger than the millimeter level, more than 6500 spacecraft have been launched into space, and the total mass of space debris has reached several thousand tons. The number of abandoned satellites has reached more than 5000 and has been increasing year by year [2]. These satellites may fall toward the Earth at any time, potentially resulting in catastrophic disasters on the ground. In 1978, a Soviet nuclear-powered satellite fell onto Canada after it collided with space debris, causing serious nuclear pollution. In 1986, an Ariane rocket exploded into 564 pieces of wreckage and 2300 small fragments, which shattered two Japanese communication satellites into pieces. In 2009, a satellite of the United States Iridium Satellite Corporation collided with a decommissioned Russian military communications satellite, generating a large amount

of space debris [3]. The probability of a catastrophic collision has been calculated as 3.7% for each vehicle in orbit, and the probability of a non-catastrophic collision is as high as 20% [3].

Various satellite internet constellation programs, such as the One Web, SpaceX Starlink, Telesat, and Hongyan constellations, will launch thousands of satellites into orbit to provide global broadband internet services, which will greatly affect the Earth's orbital resources. Space debris has become an important factor threatening space safety. The European Space Agency issues early warnings of collisions between space debris and spacecraft more than 50 times per year on average, and the number of collision avoidance maneuvers required annually has reached 22 [5]. Ensuring spacecraft safety after launch requires the orbits of space debris to be determined accurately. Compared with microwave technology for tracking space debris, laser technology is characterized by a short wavelength, small divergence angle, good directivity and monochromaticity, and strong anti-interference. A laser can be used for a real-time measurement technology with a ranging precision reaching a sub-decimeter level, which is two orders of magnitude better than the ranging precision of a microwave radar or optical telescope. The use of a laser improves the accuracy of the determination of the space debris orbit and plays an important role in enhancing spacecraft collision warning capabilities [4–6]. Many countries, including the United States, Australia, Austria, Russia, Poland, and China, have developed high-precision laser ranging technology for debris laser ranging (DLR) [7–13]. The International Laser Ranging Service [14] established a space debris laser tracking and measurement working group in 2014 to promote the research and application of high-precision space debris measurement technology and allow the high-precision monitoring and early warning of space debris in space activities, which will enhance the early warning capabilities of space targets for space situational awareness.

In DLR, the surface of the measured target diffusely reflects the incident laser beam and the return laser signal is so weak that it makes measurements difficult [11]. Improving the laser emission capability, signal receiving, and detection capabilities of the ground station is, therefore, important to DLR technologies. In recent years, DLR has advanced in terms of the pulse repetition frequency (PRF) and pulse width of the laser, single photon detection, and signal receiving. These technological advances can effectively improve the tracking capability of DLR, resulting in routine observations and applications, such as the determination of the orbits and attitudes of space debris and the verification of orbits.

As discussed above, efforts to improve the tracking ability of the laser ranging of space debris have received significant attention. The Shanghai Astronomical Observatory (SHAO), as a pioneer of DLR in China, proposed methods to improve the performances of DLR measurements, especially for small debris.

The main advances of the proposed DLR technologies are addressed in this study, including high PRF, a laser detector with low dark current noise and high efficiency, and a picosecond-pulse-width laser. Employing these technologies to the SHAO facility, we obtained a lot of DLR measurements and analyzed them to investigate DLR improvement in terms of ranging accuracy and tracking capability.

## 2. Methodology of DLR Measurements

### 2.1. Theoretical Analysis

Using the laser ranging link equation, the number of laser echo photons can be estimated from the parameters of the measuring system and the size of targets, which provides a reference for practical laser ranging. The link equation is written as [11].

$$n_0 = \frac{\lambda \eta_q}{hc} \times \frac{E_t A_r \rho S \cos \theta}{\pi \theta_t R^4} \times T^2 \times \eta_t \times \eta_r \times \alpha \qquad (1)$$

where $n_0$ is the average number of photoelectrons produced by the photodetector, $\lambda$ is the wavelength of the laser, $\eta_q$ is the quantum efficiency of the photodetector at the wavelength $\lambda$, $E_t$ is the single-pulse energy, $A_r$ is the effective receiving area of the tele-

scope, $S$ is the cross-section of the debris targets, and $\rho$ is the reflection rate of the debris targets. It is supposed that the pieces of space debris are spherical, such that $\cos\theta = 1$. $h = 6.6260693 \times 10^{-34}$ J·s is the Planck's constant, $c = 3 \times 10^8$ m/s is the speed of light, $\theta_t$ is the divergence angle of the laser beam, $R$ is the distance from the ground station to the target, $T$ is the one-way atmospheric transmittance, $\eta_t$ and $\eta_r$ are respectively the efficiencies of the transmitting and receiving optics units, and $\alpha$ is the attenuation factor. Taking into consideration the background noise ($n_1$) and detector dark noise ($n_2$), the total detecting probability of the laser signal is given by:

$$\xi = e^{-(n_1+n_2)}\left(1 - e^{-n_0}\right) = e^{-(n_1+n_2)}\left(1 - e^{-\frac{\lambda\eta_q}{hc} \times \frac{E_t A_r \rho S \cos\theta}{\pi\theta_t R^4} \times T^2 \times \eta_t \times \eta_r \times \alpha}\right) \tag{2}$$

Considering the power ($P$) and repetition rate ($f$) of the laser system, the number of echoes ($D$) from targets per second is calculated as:

$$D = fe^{-(n_1+n_2)}\left(1 - e^{-\frac{\lambda\eta_q}{hc} \times \frac{P A_r \rho S \cos\theta}{\pi\theta_t R^4} \times T^2 \times \eta_t \times \eta_r \times \alpha}\right) \tag{3}$$

In Equation (3), the number of echoes is inversely proportional to the fourth power of the distance and proportional to the cross-section, which is taken as the radar cross-section (RCS). Increasing the measuring ability requires the development of DLR technology mainly focusing on the laser system, receiving performance, detection ability, and the number of noise levels required to be decreased. Laser systems have been developed from a low repetition rate to a high repetition rate and from nanoseconds to picoseconds in terms of the pulse width. Optical and electronic developments have resulted in excellent detectors and improvements to DLR technology (i.e., a detector with a large chip and high efficiency, a low-noise avalanche photo-diode detector and a superconducting detector with super low dark noise). Additionally, the signal receiving system uses large-aperture telescopes and telescope arrays.

### 2.2. High PRF of DLR Technology

The early research on DLR technology mainly focuses on laser pulses having a low PRF and a few joules of energy to pursue high energy with a nanosecond pulse width. In 2002, Electro Optic Systems in Australia published a report on DLR technology at the 13th International Laser Ranging Conference in Washington. The report briefly introduced the research progress and measurement results of DLR technology and showed that the ranging of 15-cm space debris located at a slant range of 1250 km could be realized by the Mount Stromlo Satellite Laser Ranging station equipped with a 76-cm telescope and 532-nm nanosecond laser [8]. SHAO has actively targeted the development of international DLR technology for a satellite laser ranging (SLR) system using a receiving telescope with an aperture of 60 cm. This development allowed the laser ranging of space debris to a distance of 900 km using a high-power laser system with a PRF of 20 Hz, an average power of 40 W (at a wavelength of 532 nm), a pulse width of 40 ns in 2008, and the ranging precision at approximately 1 m [11]. The above laser generator operated in lamp pump mode with double beam synthesis and had poor beam quality and a short continuous working time. Therefore, in 2008, SHAO improved the laser ranging control system by introducing a large power laser system with high stability, a PRF of 10 Hz, an average power of 10 W (at a wavelength of 532 nm), and a pulse width of 10 ns. In 2009 and 2010, SHAO successfully conducted DLR for a rocket body and disjunction Iridium satellites at a measuring distance exceeding 1200 km, resulting in a ranging precision of 80–100 cm. In 2011, with support from the Chinese Academy of Sciences, SHAO improved the laser ranging system by upgrading the 10 W laser to a 30 W laser and conducted measurements up to 1800 km with an RCS of 3.4 m$^2$, achieving a detection success rate of more than 50% [11]. Additionally, Yunnan Astronomical Observatory of the Chinese Academy of Sciences developed a laser ranging system with a telescope having an aperture of 1.2 m

and a laser having a single pulse energy of 4.5 J at a PRF of 10 Hz and obtained successfully laser echo data of space debris [12].

(1)    The 200 Hz PRF DLR

Equation (3) shows that, for a constant number of laser echo detections, increasing the PRF of a laser can significantly reduce the laser pulse energy. Since 2013, SHAO has developed a high-power nanosecond laser system with a PRF of 200 Hz, single-pulse energy of 300 mJ, and average power of 60 W. The laser structure is shown in Figure 1. The main specifications of the laser system are a wavelength of 532 nm, a pulse width of ~10 ns, beam pointing stability of ~45 μrad, and beam quality ($M^2$) of ~3.

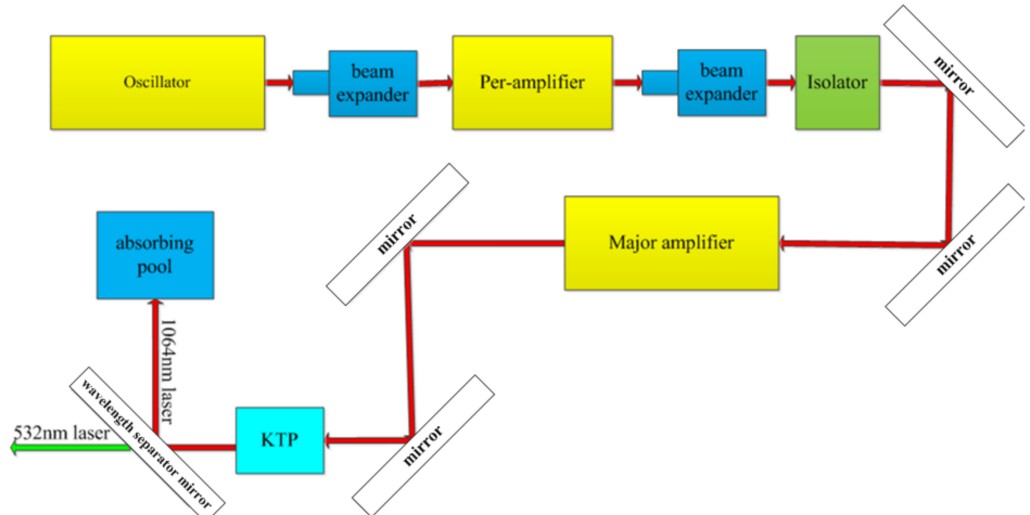

**Figure 1.** Schematic of an optical system for a high-power nanosecond laser with a PRF of 200 Hz.

The laser measurement of debris was performed by updating the range gate generator, a data acquisition unit, and data process software of the DLR system. Figure 2 shows the observation results of the DLR system at SHAO for hundreds of laps of DLR data. The x-axis and y-axis are the measured distance and the elevation of the observation, respectively. It is seen that the farthest measured distance is nearly 3000 km. The RCS is equivalent to 0.26 $m^2$ at a distance of 1000 km.

(2)    The 1 kHz PRF DLR

In 2015, Graz Station in Austria used a 532 nm laser with a PRF of 1 kHz, a pulse width of 10 ns, and single-pulse energy of 25 mJ and a telescope with a receiving aperture of 50 cm to realize DLR for an orbit height of 3000 km and a minimum area of 0.3 $m^2$ [9].

SHAO also updated a picosecond laser system adopting a dual-pulse mode, power of 4.2 W, a wavelength of 532 nm, single pulse energy of 2.1 mJ, and PRF of 1 kHz. The RCS of the measured target was 2 to 12 $m^2$ and the ranging precision was at a sub-decimeter level [15]. The range residual data of the debris target (ID 38346) show a visible signature effect, marked by boxes in Figure 3. This confirms that the attitude of debris targets changes and the laser ranging techniques are suitable for measuring the attitude of targets.

(3)    The 100 kHz PRF laser ranging

A high PRF of laser ranging provides a high echo data density and allows a fast target search. It means that DLR makes lower laser energy available by increasing the PRF of laser ranging, to keep tracking capabilities. Adopting a high PRF for laser ranging systems is an important technical approach to improve detection capabilities and will be a trend in future technological development. SLR technology has surpassed a PRF of 100 kHz [16–19], laying the foundation for an ultra-high DLR repetition rate of 100 kHz–1 MHz. Additionally, SHAO has developed laser ranging technology for an ultra-high PRF of 100 kHz.

**The measured distance vs. elevation of debris targets**

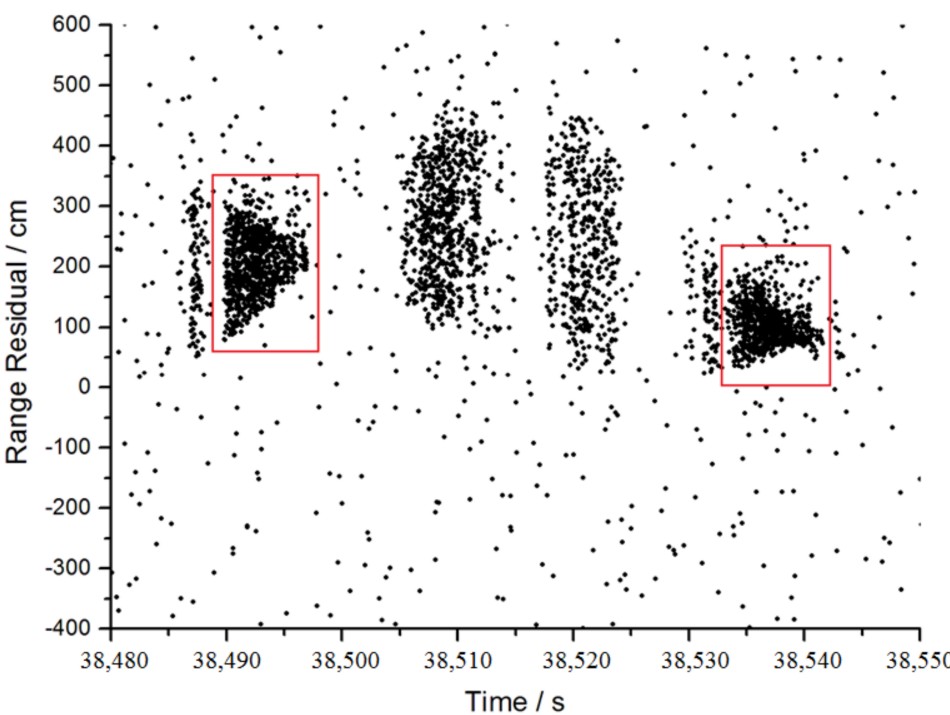

**Figure 2.** Laser ranging results using a 60 W laser with the SHAO laser ranging system.

**Target: H-2A DEB (ID: 38346)**

**Figure 3.** Laser ranging of the debris target (ID 38346) with a dual-pulse picosecond 532 nm laser at a repetition rate of 1 kHz and power of 4.2 W.

The ultra-high PRF of laser ranging causes a much higher intensity of the laser atmospheric back-scattered signal, which is likely to damage the single-photon detector.

Therefore, a method of alternately transmitting and receiving laser pulses has been adopted for the ultra-high PRF of laser ranging. In this method, when the laser light is emitted, the detector does not work until the laser echo returns. This laser ranging mode separates the back-scattered signal from the laser echo in time and avoids the interference of the back-scattered signal.

The procedure of the above mode is shown in Figure 4, where $I_p$ is the time interval between laser pulses within the group (5 μs for 200 kHz), on/off refers to the working status of the laser and detector, and *ToF* is the flight time of the laser from the ground to the target. The working time length of the alternating mode is calculated according to *ToF*, and the number of laser pulses within the time of flight is calculated as:

$$N = \frac{ToF}{I_P} \tag{4}$$

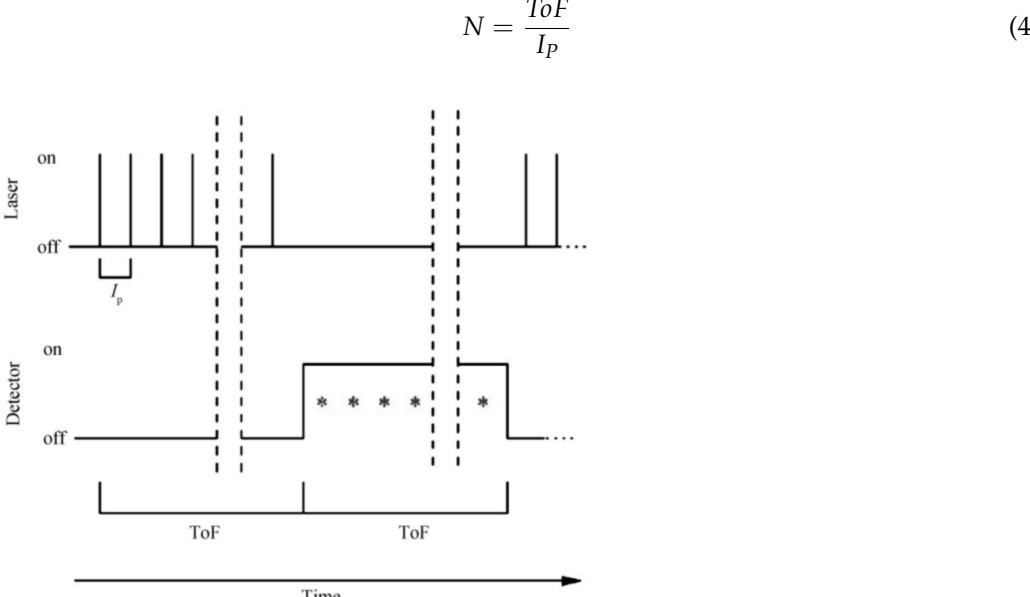

**Figure 4.** Mode of laser emission and reception alternation for laser ranging at an ultra-high PRF.

The laser emission and reception are processed alternately to perform 100 kHz SLR measurements.

By updating data acquisition software and range gate generator, 100 kHz laser ranging was performed to track satellites with a laser retroreflector using the SHAO laser ranging system. Figure 5 shows the measurements of the Sentinel-3B satellite at a PRF of 100 kHz with single pulse energy of 40 uJ in wavelength of 532 nm. The range residuals for 100 kHz measurements are plotted in Figure 5, where the laser echo signals appear periodically because of laser transmitting and receiving alternately. In the next step, we have a plan to apply 100 kHz SLR technology to DLR measurements and develop 500 kHz laser ranging technology.

### 2.3. Low Noise and High Efficient Laser Echo Detection

For the detection of laser signals of cooperative targets with laser reflectors, a single-photon avalanche diode (SPAD) detector with single-photon sensitivity, timing accuracy of 30–50 ps, a chip diameter of 200 μm, and a quantum detection efficiency of 20% is widely used at laser ranging stations [20]. The photon detector has a relatively high level of dark noise. In the high-repetition-rate working mode, the dark noise reaches approximately 100 kHz. Combined with the effect of background noise, the detector noise reaches a level of hundreds of kiloherz, which may trigger the avalanche process and cause false detection.

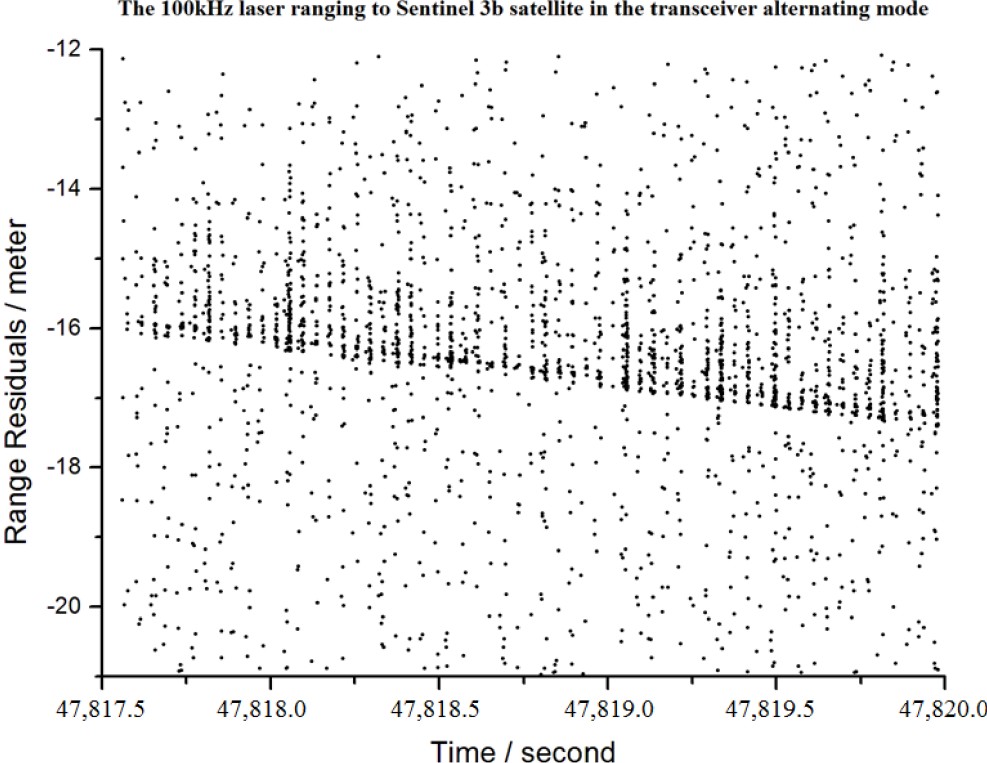

**Figure 5.** Laser ranging of the Sentinel 3B satellite at a PRF of 100 kHz.

A wide range gate and low noise level of laser detection are required to handle the large-ranging error coming from poor accuracy of debris orbit prediction. The avalanche photodiode (APD) detector technology with a low level of dark noise, high efficiency (>40%), and high sensitivity have been developed [21]. Additionally, a large chip is required to ensure that the detector has a sufficient optical receiving field of view. The APD detector in the Geiger mode operation works at a voltage higher than the breakdown voltage to realize sufficient gain, after the photons are absorbed, the detector can quickly avalanche output to meet the requirements of single-photon detection and realize higher detection efficiency and sensitivity, as shown in Figure 6.

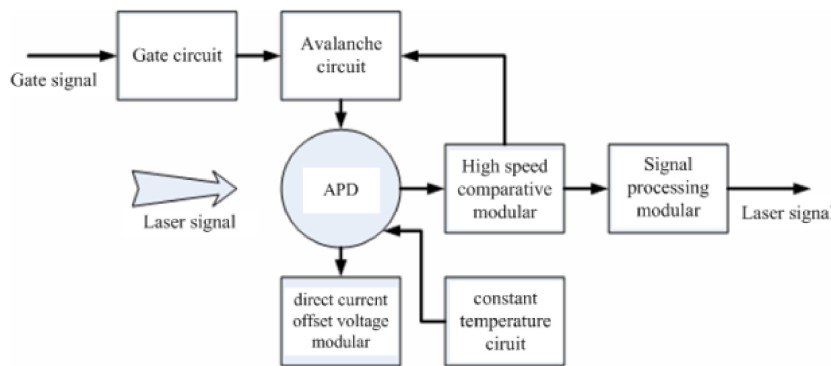

**Figure 6.** Principle of the avalanche photodiode detector for DLR measurements.

SHAO developed a low-noise, highly efficient single-photon detector for DLR measurements. The main specifications of the detector are a diameter of the photosensitive surface of 500 μm, dark noise of approximately is less than 1 kHz at a PRF of 1 kHz, detection efficiency exceeding 40% at a wavelength of 532 nm, and detection accuracy of approximately 500 ps.

The laser echo and noise signal of the APD and SPAD detectors were compared for the laser tracking of satellites using the same system parameters. Table 1 shows that the APD detector outperforms the SPAD detector in terms of detection efficiency and noise signal. The APD detector is thus suitable for the detection of weak laser echo signals from debris targets.

**Table 1.** Statistics of laser echoes and signal ratio for APD and SPAD detectors.

| Type | Total Points | Validated Points | Signal Ratio/% | Noise Ratio/% |
|------|--------------|------------------|----------------|---------------|
| APD | 892 | 781 | 19.74 | 12.44 |
| SPAD | 250 | 101 | 7.763 | 59.6 |

The superconducting nanowire single-photon detector (SNSPD) has become the most competitive detector among detectors developed in past decades. It has excellent performance, such as a wide spectral response range, ultra-high detection efficiency, extremely low dark count rate, and little time jitter. The SNSPD is expected to provide breakthroughs in space target detection capabilities and help to improve the search capabilities and acquisition speed for space targets.

Since 2015, SHAO has cooperated with the Shanghai Institute of Microsystem and Information Technology to develop the first SNSPD for high-precision laser ranging [22]. This detector has a photosensitive surface diameter of 100 μm, the detection efficiency was up to 60% with a bias current of 28 uA as shown in Figure 7b, the dark noise (DCR) was less than 10 Hz, and also the time jitter was 68 ps, as shown in Figure 7b. The laser echo signals received in the 200-μm-diameter core of the fiber are focused on the SNSPD by an aspheric lens.

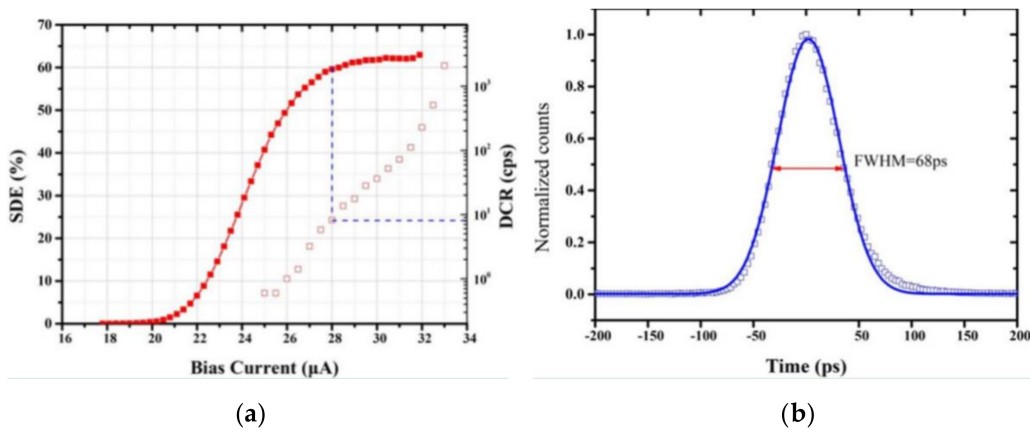

(a)                             (b)

**Figure 7.** Detection efficiency: (**a**) dark noise, (**b**) time jitter of the SNSPD.

A five-dimensional tunable fiber-coupled aspheric lens realizes the highly efficient coupling of light to the narrow 200-μm-diameter core under the highly dynamic circumstances of the telescope. Figure 8 illustrates the experimental configuration of the efficient coupling and application of the SNSPD in laser ranging.

SHAO applied the SNSPD detector to the centimeter-level-precision laser ranging of satellites and sub-decimeter-level-precision laser ranging of debris targets. Figure 9 shows the laser range residuals for the debris of the CZ_2C rocket, which is characterized by less noise and an obvious laser signal. The ratio of signal to noise was up to 1.9, higher than that of the APD detector, as shown in Table 1. This result demonstrates the advantage of the high signal-to-noise ratio for the SNSPD detector.

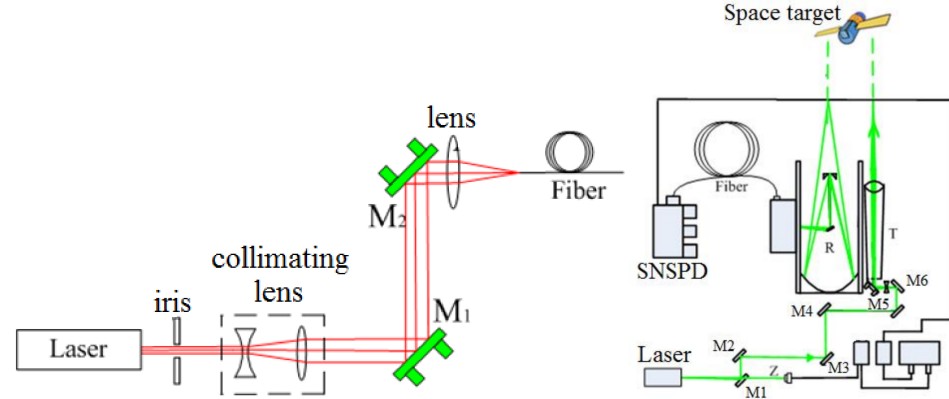

**Figure 8.** Experimental configuration of the efficient coupling of light to the fiber (**left**) and application of the SNSPD in laser ranging (**right**).

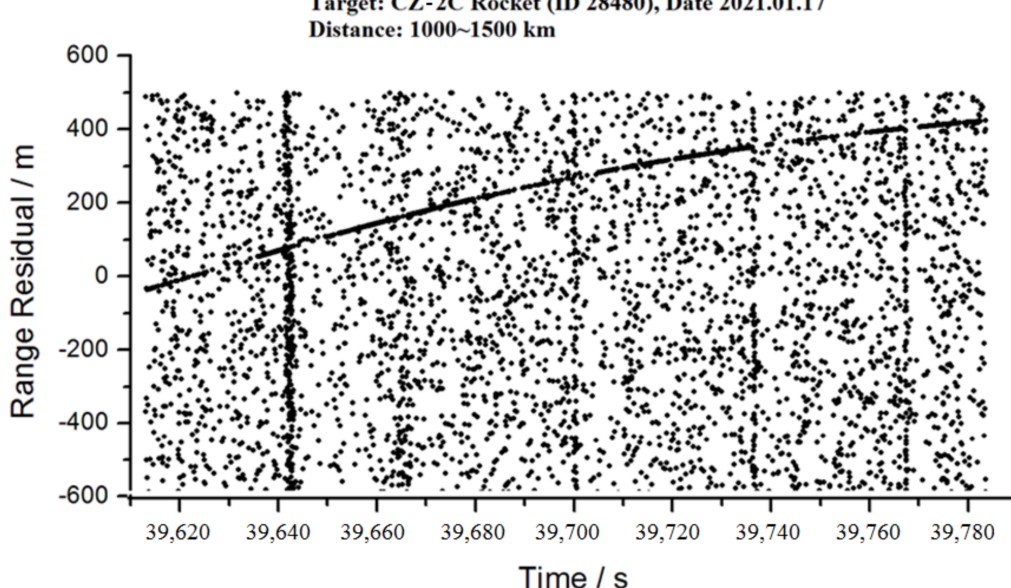

**Figure 9.** Measurement results of DLR using the SNSPD detector at a PRF of 1 kHz with a high ratio of signal to noise.

### 2.4. Large-Aperture Telescope for DLR Measurements

A larger telescope aperture can receive more laser photon echoes and then realize a better performance of DLR. In 1994, at the Ninth International Laser Ranging Conference in Canberra, FUGATE R announced that a telescope with an aperture of 3.5 m could range space debris at a distance of 1000 km. In 2004, Electro Optic Systems in Australia reconstructed a DLR telescope having an aperture of 1.8 m and ranged space debris of a size of 10 cm at a distance of 1000 km using an average laser power of 100 W [1]. The system could carry out the three-dimensional measurement of space debris, realizing the high-precision positioning and orbit determination of debris targets, and improve the accuracy of cataloging space debris. In 2012, the Grasse station in France employed a telescope with an aperture of 1.56 m to achieve DLR for an orbital height of 1700 km [23].

SHAO developed a DLR system with a telescope having an aperture of 1.8 m and a laser system having a PRF of 200 Hz, pulse energy of 300 mJ and a wavelength of 532 nm. However, the noise from the background and debris targets was several times greater than that of the SHAO's DLR system with a telescope having an aperture of 60 cm, which affects the laser echo detection and then deteriorates the detection capability. Narrow spectral filters are commonly used to reduce the level of background noise in

a laser measurement system by using the homochromy of the laser signal (e.g., a high-efficiency narrow-bandwidth spectral filter). Figure 10 shows the transmittance of a narrow-bandwidth spectral filter. The main characteristics of the filter are a central wavelength of 532 nm, a bandwidth of ±1 nm, and a transmittance of greater than 90% at a wavelength of 532 nm.

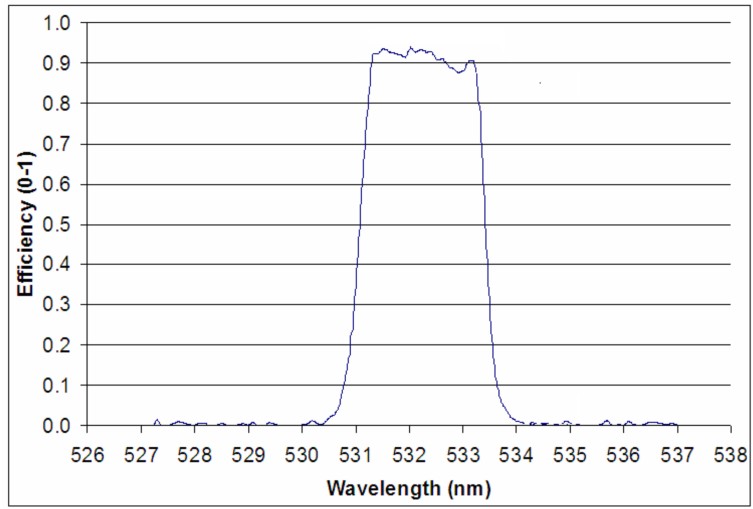

**Figure 10.** Transmittance of a narrow bandwidth spectral filter.

The technologies of the DLR measurements were implemented at the SHAO, which enable ranging measurements of small and far space debris at a distance of 1700 km (with an RCS of 0.05 m², catalog ID 1520) and at a distance exceeding 6700 km (with an RCS of 13.6 m², catalog ID 28118), as shown in Figures 11 and 12, respectively. These observations mark the first time that debris targets have been measured up to a distance of 6700 km.

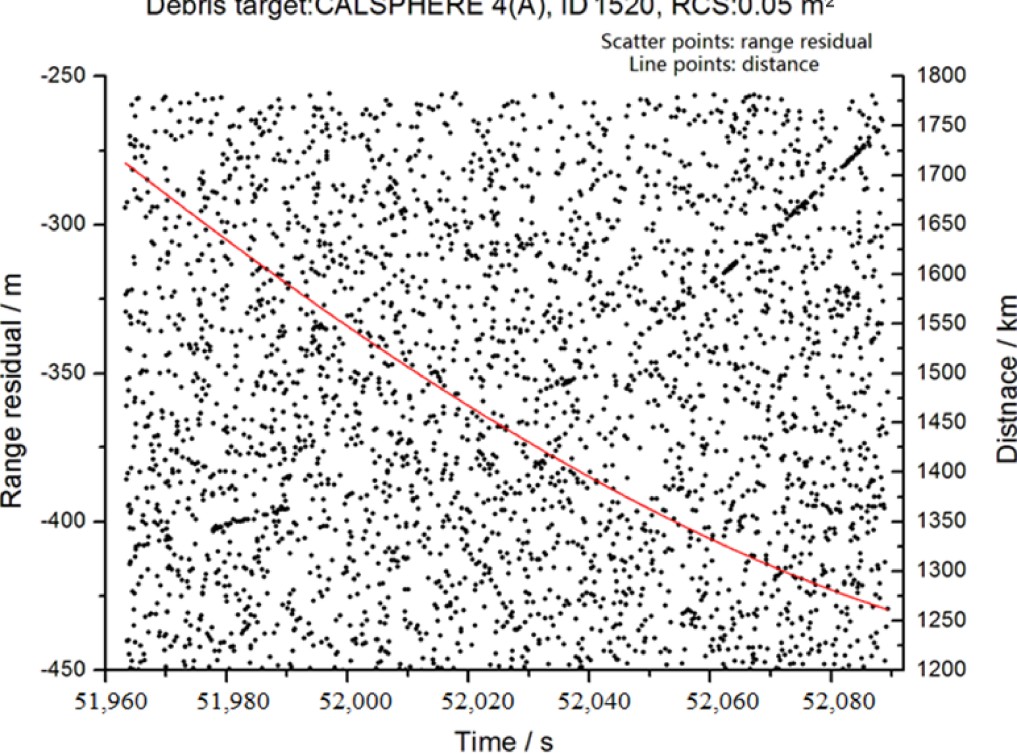

**Figure 11.** DLR measurements of a target with an RCS of 0.05 m² (catalog ID 1520).

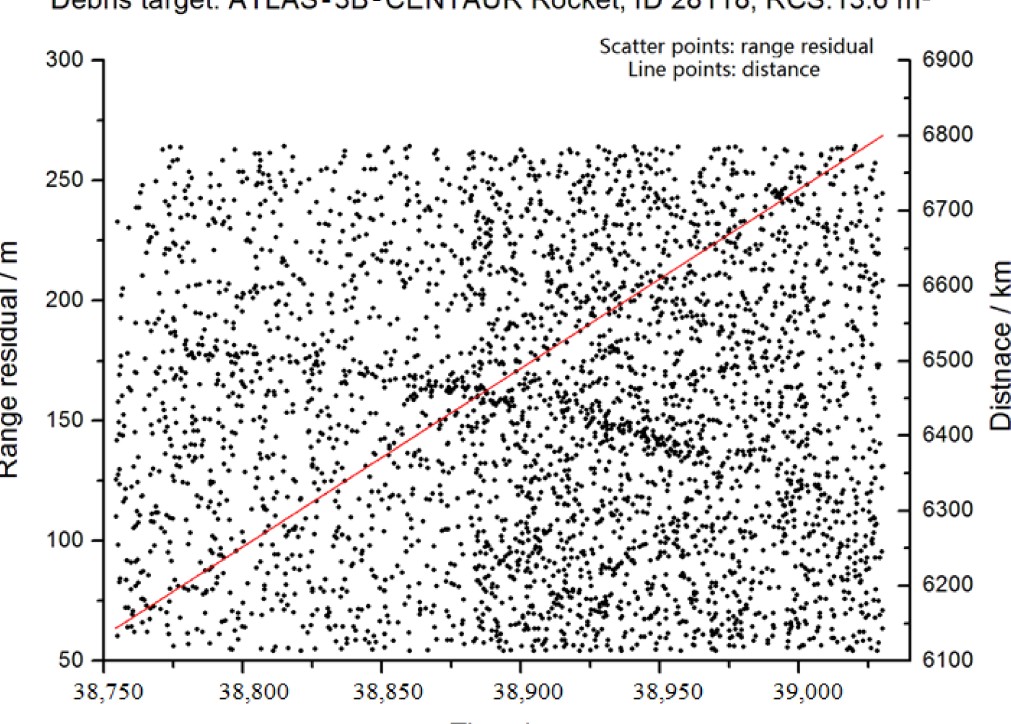

**Figure 12.** DLR measurements of a target at a distance exceeding 6700 km (catalog ID 28118).

### 2.5. Telescope Array and Multi-Static Laser Ranging Measurements

In laser ranging measurements, the laser echoes cover a certain area on the ground, and they can thus be received and detected by an array of several or more telescopes on the ground, which causes increases in the receiving area. These telescopes receive and detect laser echo signals at the same time, significantly increasing the number of laser echoes received by the ground station system per unit time, which is equivalent to improving the receiving capability of a single large-aperture telescope, as shown in Figure 13. The main telescope in the array actively emits laser signals to the target, whereas all the telescopes of the array receive the echo signals.

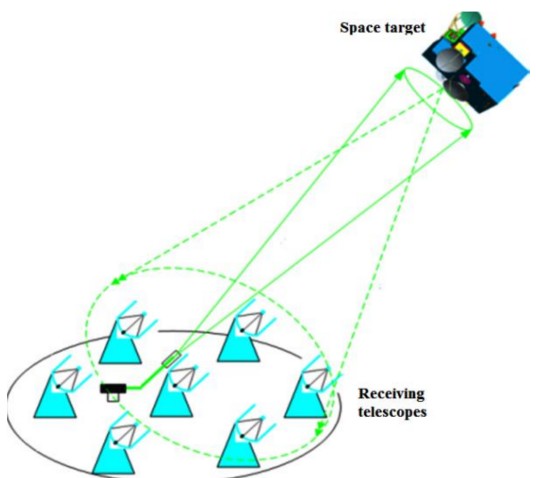

**Figure 13.** The receiving and measurement of laser echo signals by a telescope array.

The detection probability and receiving capability of a ranging system consisting of multiple telescopes receiving signals has been studied using dual telescopes that have apertures of 1.56 m and 60 cm and are separated by 60 m to receive laser echo signals [24,25]. The 60 cm telescope emitted a laser signal to the targets and the two telescopes received an echo signal at the same time.

Using the clock of a global navigation satellite system to solve the time synchronization of the two separate telescope systems and considering the actual transmission path of a laser signal, the range gate control of the remote receiving telescope has been realized for laser ranging measurements. Figure 14 illustrates the multi-static measurements of a telescope array. The range gate value is $\Delta\tau_0 = \tau_{u0} + \tau_{d0}$ for the main telescope (#0), $\Delta\tau_1 = \tau_{u1} + \tau_{d1}$ for the fellow receiving following telescope (#1), $\Delta\tau_2 = \tau_{u2} + \tau_{d2}$ for the fellow receiving telescope (#2), respectively. Considering the negligible differences in $\tau_{u0}$, $\tau_{u1}$, and $\tau_{u2}$ for the same target, the range gate values can be calculated based on the laser-signal uplink time from the main telescope and downlink time to the respective following telescopes.

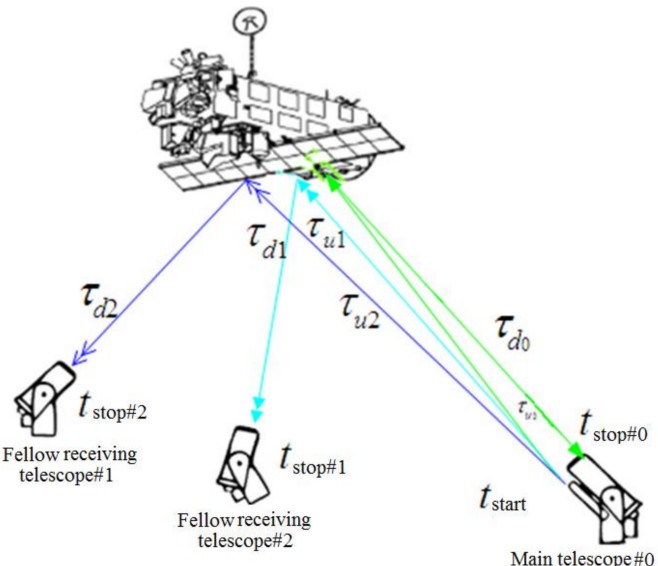

**Figure 14.** Multi-static laser-ranging measurements of a telescope array.

Figure 15 shows the real-time combination of laser range residual data recorded by the 1.56 m and 60 cm telescopes, respectively, which increases the total number of laser echoes.

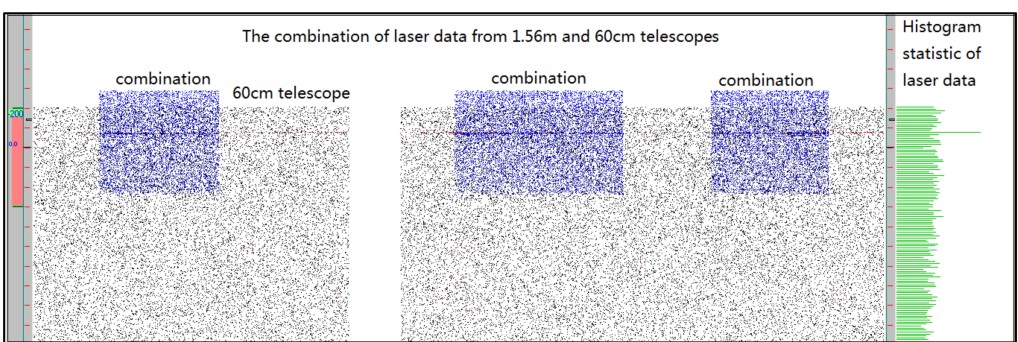

**Figure 15.** Real-time combination of laser data from the 1.56 m and 60 cm telescopes.

Figure 16 shows the DLR measurements for the 1.56 m and 60 cm telescopes, respectively. Compared with the telescope having an aperture of 60 cm alone, the total number of laser echoes received by the two telescopes together has increased by a factor of 4–5.

Consequently, the receiving ability of the two telescopes is equivalent to that of a telescope system with an aperture of approximately 1.61 m. This result verifies the feasibility and technical advantages of multiple telescopes receiving signals.

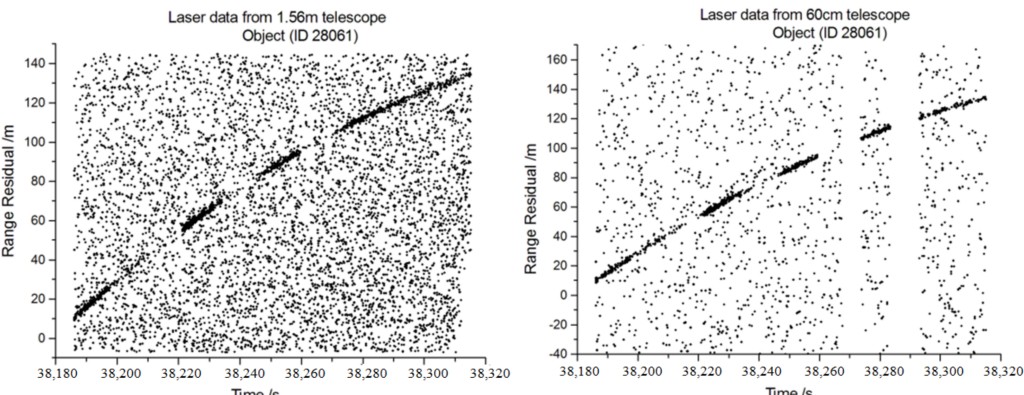

**Figure 16.** DLR measurements were obtained using the 1.56 m and 60 cm telescopes.

SHAO has two laser-ranging telescopes with a receiving aperture of 60 cm, separated by 2.5 km, each having a receiving aperture of 60 cm [26]. The main telescope system located at the top of Sheshan Mountain has a laser emitter and receives echo signals at the same time. The fellow telescope located in a science park only receives laser echo signals.

DLR measurements were successfully achieved using the dual telescopes. The measurement results of the debris targets given in Table 2 show that the measuring precision of laser ranging of the dual telescopes is comparable. It is concluded for this comparability that the precise control of range gate synchronization is available for the remote receiving telescope system and the detection of debris laser signals can be improved by a long-distance multi-static telescope laser ranging system.

**Table 2.** Results of debris measurements obtained using the dual telescopes.

| ID | Space Debris | RCS: m$^2$ | Measured Distance (km) | Ranging Precision (cm) | |
|---|---|---|---|---|---|
| | | | | Main Telescope | Fellow Telescope |
| 21,610 | ArianeR_B | 16.84 | 1390 | 129.2 | 149 |
| 16,182 | SL-16R_B | 12.05 | 1030 | 141.0 | 141.7 |
| 11,672 | SL-14R_B | 4.21 | 1160 | 56.2 | 64.7 |
| 16,720 | SL-14R_B | 3.58 | 1170 | 74.9 | 62.2 |

The approach is also allowable for debris targets with a common view in telescopes hundreds of kilometers away, which can make full use of the laser ranging system and geometric advantages of each station.

## 3. Characteristic of Picosecond Laser Measurements for Debris Targets

### 3.1. Picosecond Laser Transmission

The peak power of a pulse laser system is given by:

$$P_{peak} = \frac{P_0}{\tau f} \tag{5}$$

where $P_{peak}$ is the peak power of the laser, $P_0$ is the average power of the laser, $\tau$ is the pulse width, and $f$ is the repetition rate of the laser.

For the previously mentioned picosecond laser (4.2 W, 1 kHz, and a pulse of 30 ps), the peak power is as high as 70 MW. The diameter of the laser beam emitted by the transmitting

telescope is approximately 8 cm, and the peak power density is calculated as 1.39 MW/cm$^2$ according to the following equation:

$$P_{peak\ power} = \frac{P_{peak}}{\pi(d/2)} \tag{6}$$

When such a high-intensity laser beam propagates through the atmosphere, there is not only a nonlinear effect but also the refractive index of the medium changes according to the following equation [27]:

$$n = n_0 + n_2 I(r) \tag{7}$$

where $n$ is the changed refractive index, $n_0$ is the refractive index without the laser, $n_2$ is the coefficient of the refractive index change, and $I(r)$ is the laser intensity along the transmission path. For the Gaussian distribution laser beam, the laser energy is high at the center and weak at the side. Strong laser transmission in the atmosphere forms a lentoid that would allow easier self-focusing, and the angle of laser beam divergence would be at the diffraction limit in this case. The diffraction limit is expressed as:

$$r = M^2 \frac{2\lambda R}{\pi d} \tag{8}$$

where, $M^2$ is the laser beam quality, $\lambda$ is the laser wavelength, $R$ is the distance of laser transmission, $d$ is the diameter of the laser spot. For the previously mentioned picosecond laser ($M^2 = 1.2$, $\lambda = 532$ nm, $R = 1000$ km, $d = 8$ cm), the divergence angle ($\theta$) of diffraction limit is approximately 4.15 μrad by:

$$\theta = \frac{r - d}{R} \tag{9}$$

This divergence angle ($\theta$) is much less than the one at the transmitting telescope. According to the laser ranging link equation, the number of laser echo photons increases in inverse proportion to $\theta^2$. For the previously mentioned nanosecond laser (60 W, 200 Hz, and a pulse width of 8 ns), the peak power density is approximately 0.74 MW/cm$^2$, which is lower than that of the 4.2 W picosecond laser system, consequently, its nonlinear effect is weak. Figure 17 shows the measurements for the 60 W nanosecond laser (left) and 4.2 W picosecond laser (right). The x-axis and y-axis present the RCS of targets and the measured range of the targets, respectively.

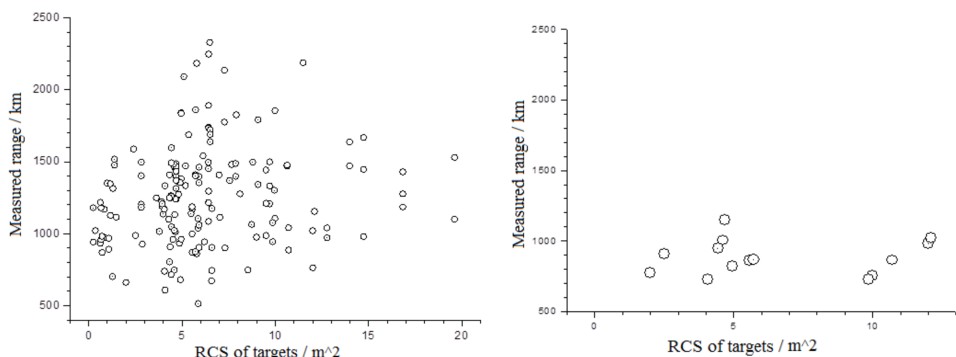

**Figure 17.** Results of the laser ranging of debris using the 60-W (200-Hz) nanosecond laser (**left**) and 4.2-W (1-kHz) picosecond laser (**right**).

Figure 17 shows that the longest ranging distance is 1150 km (with an RCS of around 5 m$^2$) for the 4.2 W picosecond laser and 2100 km (with an RCS of 5 m$^2$) for the 60 W nanosecond laser. Additionally, the normalized measuring capability of the 4.2 W picosecond laser is 6.4 times higher than the 60 W nanosecond laser. These results validate that the self-focusing of the high-peak-power picosecond laser signal is easier when transmitting at

the diffraction limit angle in the atmosphere. The greater number of laser echo photons is beneficial for the telescope to receive the laser echoes.

### 3.2. Pulse-Bursts Picosecond Laser System with High Power

For the picosecond laser with a PRF of 1 kHz, it have high peak power, but the average power and single-pulse energy cannot be increased easily. Therefore, a pulse-burst picosecond laser was developed by SHAO for the DLR to achieve high power and large single-pulse energy. A Michelson interferometer was used to achieve narrowly spaced pulse bursts, as shown in Figure 18 (left). BS1, BS2, and BS3 denote beam splitters with transmission rates of 50% whereas M1, M2, M3, and M4 denote zero-degree total reflectors coated at a wavelength of 1064 nm. Each zero-degree total reflector corresponds to a pulse generator in a burst. Pulses in bursts with equal spacing $\Delta t$ were obtained by increasing the lengths of M1 to BS1, M2 to BS1, M3 to BS1, M4 to BS1 to $\Delta L$ [28,29]. Here, we took the length $\Delta L = 150$ mm and a pulse spacing of $\Delta t = 2 \times \Delta L / c = 1$ ns (where $c$ is the speed of light). Pulses of a mode-locked picosecond laser with a PRF of 80 MHz are divided into four pulses in Figure 18 (right).

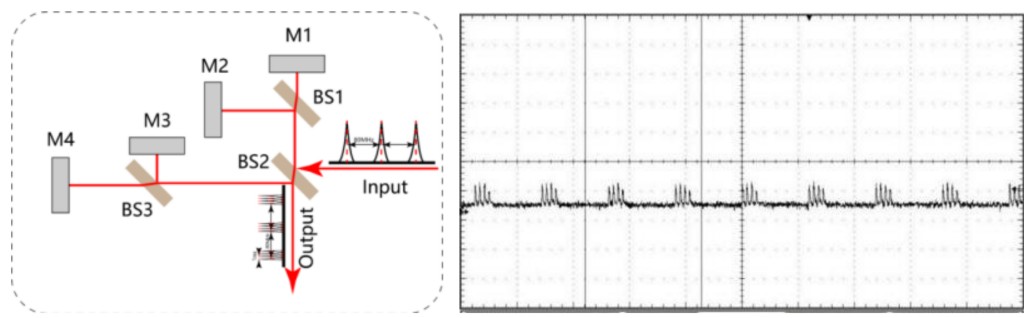

**Figure 18.** Schematic diagram of a Michelson interferometer for pulse bursts (**left**) and the waveform of the pulse bursts (**right**).

To realize high-power lasers with picosecond pulse bursts, a burst-mode regenerative amplifier (RA) and a two-stage traveling-wave amplification have been adopted for the amplifier system [30,31], as shown in Figure 19. The Nd:YAG RA, which acts as a pre-amplifier, reduces the PRF of the seed laser from 80 MHz to 1 kHz through electro-optic modulation. The seed pulse burst laser is injected into the RA through the optical isolation system from P1 and then comes out from P1 after the amplification. The Faraday rotator (FR) and half-wave plate (HWP$_1$) act as an optical isolation system that avoids the effect of the output amplified beam returning to the seed oscillation cavity. The quarter-wave plate (QWP) and Pockels cell (PC) create the pulse.

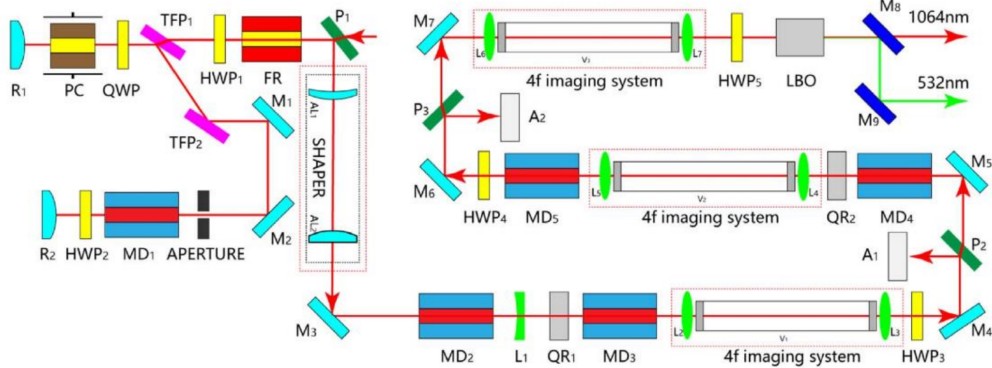

**Figure 19.** Schematic diagram of the amplifier and second harmonic generation construction of a pulse-burst picosecond laser system.

To maintain the beam quality during the two stages single pass double module amplification, two 4f imaging systems are used, and all subsequent imaging systems contain vacuum tubes to prevent air breakdown. A half-wave plate and 90-degree quartz rotator help to compensate for the difference in optical power between the tangential and radial polarization components of the thermal lens. Two 45-degree polarizers are used to improve the degree of polarization. A frequency conversion module, which is used to adjust the beam size and divergence, is placed behind the 4f imaging system to maximize the conversion efficiency. In further improving the conversion efficiency, a 1064 nm thin-film polarizer is used to purify the polarization direction of the amplified fundamental laser. After the Nd:YAG RA and two-stage traveling-wave amplification, the seed pulse-burst is amplified to 40 W at a wavelength of 1064 nm and a PRF of 1 kHz.

The second harmonic waves are generated by the lithium triborate (LBO) crystal, which was chosen for its high damage threshold, relatively high nonlinearity, large acceptance angle, and no spatial walk-off for type-I noncritical phase matching at 1064 nm. The crystal is combined with a half-wave plate to realize the best phase matching condition. Owing to the configuration and technologies addressed above, the pulse-bursts picosecond laser with high power was achieved (i.e., an average power of 20 W at 532 nm and frequency doubling efficiency of 50%).

### 3.3. Pulse-Bursts Picosecond Laser Ranging to Debris Targets

DLR measurements were performed applying the above pulse-burst picosecond laser system to the existing DLR system of SHAO. Figure 20 shows the measurements of the Long March CZ-4 debris target (ID 20853, RCS 0.6 $m^2$). The debris ranging precision is 16.4 cm, which gradually improves as the size of the debris decreases. Among the measurements, the longest ranging distance is 1726.8 km (with an RCS of 0.91 $m^2$, which is equivalent to an RCS of 0.1 $m^2$ at a distance of 1000 km). Figure 21 shows the good detection ability, with the x-axis is the RCS of targets and the y-axis is the measured range and precision of the targets. Employing the pulse-burst mode, the average power of the laser system can be increased by summing the number of laser pluses. This increase in power will allow the long-distance ranging of small debris in the future.

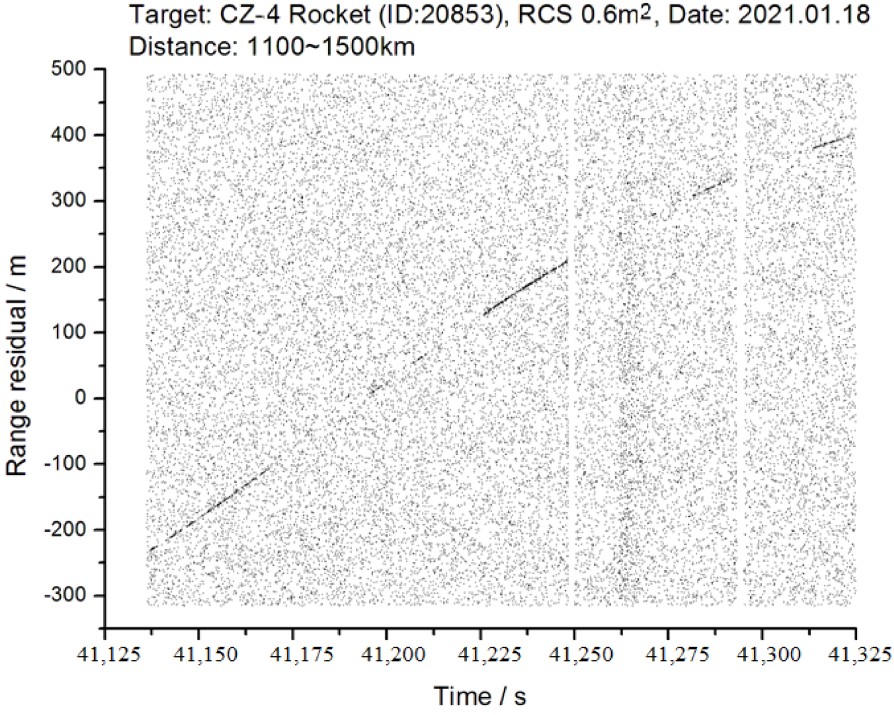

**Figure 20.** Measurements of the debris target using the pulse-burst picosecond laser.

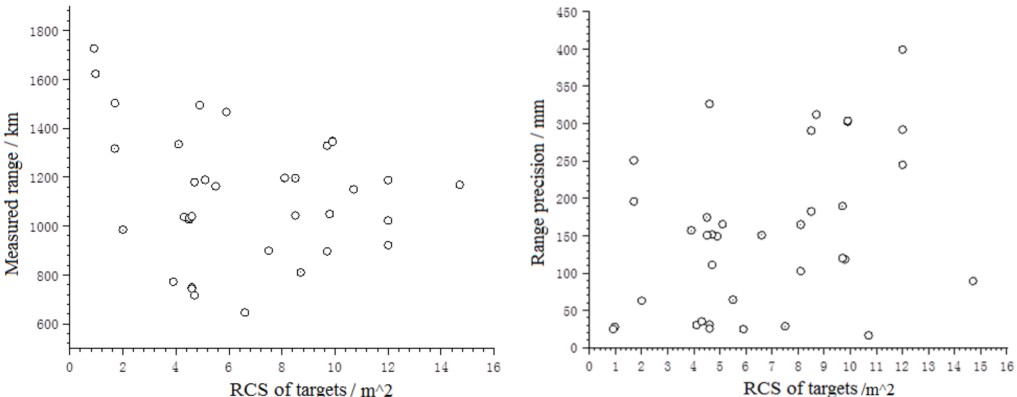

**Figure 21.** Results of the measured range (**left**) and precision (**right**) vs. the RCS obtained using the pulse-burst picosecond laser.

## 4. Conclusions

This paper presented the developments in the research and implementation of DLR technologies. The detection capabilities of the DLR system have been improved by increasing the PRF of laser ranging and reducing background light noise and dark noise in the detector. For the nanosecond laser with an average power of 60 W, a PRF of 200 Hz, and a wavelength of 532 nm, the measurement performance was equivalent to an RCS of 0.26 $m^2$ at a distance of 1000 km. Meanwhile, using a 532 nm pulse-burst picosecond laser with an average power of 20 W and a PRF of 1 kHz, debris measurements was achievable up to 1726.8 km (with an RCS of 0.91 $m^2$, corresponding to an RCS of 0.1 $m^2$ at a distance of 1000 km). It was demonstrated that the picosecond laser outperforms the nanosecond laser in terms of ranging precision and capabilities, and it is also expected to be a new direction of DLR development. Owing to the ultra-high repetition rate of SLR technology, an increase in the single-pulse energy of the laser effectively increases the average number of echoes per unit time. Furthermore, the development of laser technology with a PRF of 100 to 500 kHz and a transceiver alternating mode will provide a new way of making ranging measurements.

**Author Contributions:** H.Z. designed the technology of the debris laser ranging (DLR) system; M.L. was in charge of the laser system and analyzed the characteristics of the pico-laser; H.D. updated the laser system and observations; S.C. and J.S. performed the experiments; Z.W. designed the telescope array of laser ranging; Z.Z. and A.Z. designed the technology of the DLR system; H.Z. and M.L. analyzed the data; S.C. and J.S. wrote the draft of the paper; and H.Z. and M.L. modified the paper. All authors have read and agreed to the published version of the manuscript.

**Funding:** This work was supported by the National Natural Science Foundation of China (12003056 and 11903066), Natural Science Foundation of Shanghai (20ZR1467500), and the Key Research Program of the Chinese Academy of Sciences (ZDRW-KT-2019-3-6).

**Institutional Review Board Statement:** Not applicable.

**Informed Consent Statement:** Not applicable.

**Acknowledgments:** The authors thank the Space Track Organization (www.space-track.org (accessed on 23 September 2021)) for making available the two-line elements and information of the cataloged objects; Institute of Applied Electronics of CAEP and Beijing University of Technology for providing the laser system; and East China Normal University and Shanghai Institute of Microsystem and Information Technology of CAS for providing the laser detector.

**Conflicts of Interest:** The authors declare no conflict of interest.

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
