# Peer review of "Developments of Space Debris Laser Ranging Technology Including the Applications of Picosecond Lasers"

_applsci, doi:10.3390/app112110080_

Round 1

Reviewer 1 Report

The paper by Haifeng Zhang et al. reports on “Developments of Space Debris Laser Ranging Technology and it Applied with Picosecond Laser”. The paper summarizes recent achievements in the field of satellite and debris laser ranging at the Shanghai Astronomical Observatory.

The paper covers many important aspects of progress in the field including progress toward ranging to space debris with small cross sections. I guess the paper will be of interest for the international laser ranging community and the community interested in the development of the space debris problem.

Before the paper is published the following points should be addressed:

The title should be corrected to “ Developments of Space Debris Laser Ranging Technology Including the Application of Picosecond Lasers” or similar, since “and it Applied with” is unclear.

“Sectio” should be corrected in the acstract.

In Figure 5 the x- and y-axis labeling is missing. For the 100 kHz PRF laser system the average power or pulse energy is not given.

In Figure 7 the figure captin should refer to the graphs and explain the quantities shown in the graph. How is the timing jitter related to the figure on the right.

Page 9: What is a “five-dimensional tunable fiber-coupled aspheric lens”?

Figure 9: The y-axis (range residuals) should not extend into the lower part of the graph signal-to-noise ration).

Some of the references are incorrect, e.g..

Ref 17: D. Hampf, E. Schafer, F. Sproll, et al. CEAS Space 11, 363-370 (2019)

Ref 18: D. Hampf, P. Wagner, E. Schafer, et al. …

Author Response

Thank you very much for your comments to our manuscript. And thank you very much for pointing the mistake, we will revise it in our revision as you say and try our best to make it good for publication. We have highlighted the changes in our manuscript within the document by using colored text in our manuscript. The response to the  comments was shown in the attachment  by using colored text.

Reviewer 2 Report

The authors present in this paper several configurations for Debris Laser Ranging techniques. The title suggests as the main topic of the paper the picosecond laser pluses used for DLR. However, only a small part of this manuscript is dedicated to picosecond pulses. From this point of view, the title is not in agreement with content of the manuscript. Several techniques are presented in the same manuscript. Probably focusing on only one technique, the one suggested by the title using the picosecond laser, with more details, measurements and performances could give a much better paper showing a degree of novelty. Also, a review of different techniques developed by authors could be interesting for the readers, but the title has to be adapted accordingly.

The word “technology” used is probably not the proper one. I consider that the term “technique” is more appropriate.

Some checks and modifications of the text are needed as mentioned below.

Abstract:

“This paper report DLR technologies of the high pulse repetition frequency (PRF) laser pulse,large-aperture telescope,telescope array , multi-static station, and so on.” The abstract has to be concise and clear. The expression “… and so on.” is not very appropriate. Please be more specific on what the paper reports. I will recommend to put more in evidence from the beginning of the abstract the original outcome of this research in relation with the title.

Page 4. Figure 1. The last mirrors after KTP should be a wavelength separator mirror, not “optical spectroscope” as market on picture.

Page 7. Please explain more in detail the figure 5. There is no axis. What this plot represents?

Page 11. A distance of 6700 km is mentioned. However, there is no information about the technical details of this measurement in comparison with the previously presented data in terms of laser parameters or detection device. Also, there is no reference provided where to find such details. If the authors consider to include this information, then some reference or more data should be mentioned.

Page 16, row 423. The laser pulses are selected, but not created by a Pockels cell.

A comments could be probably added about the daylight effects on debris laser ranging performances as additional noise to be accounted.

Please correct some typing errors:

Row 12: Place a space after the end of sentence: “…debris. This…”
Row 13: Place a space after comma “…laser pulse, large-aperture telescope, telescope…”
Row 14: Remove space before comma “…array, multi-static…”

Row 168: In the sentence “This confirms that the attitude of debris targets”, please verify: attitude (as behaviour), or altitude?

Author Response

(The authors gave the same response as above.)

Round 2

Reviewer 2 Report

Before the final version, I will suggest to the authors to consider some text editing rules and corrections such as:

- a space has to be places between a number and a unit symbol.

Examples:
row 251: correct 10 Hz; not correct 10Hz
row 203: correct 532 nm; not correct 532nm

- There is always a space between a number and the symbol %: 

row 250: correct 60 %, not correct 60%

- for micron subunits use the symbol μ:

row 203: 40 μJ , not 40uJ
row 250: 28 μA, not 28uA

Author Response

Thank you very much for your comments to our manuscript. And thank you very much for pointing the mistake, we have revised it in our revision as you say and try our best to make it good for publication. We have highlighted the changes in our manuscript within the document by using colored text in our manuscript.

This manuscript is a resubmission of an earlier submission. The following is a list of the peer review reports and author responses from that submission.

Round 1

Reviewer 1 Report

The authors have made significant changes in the language and writing style, of the manuscript. But it still looks like a technical report and needs major improvements.

The main concern here is whats innovation. The scientific motivation for the proposed technological advancements for DLR at SHAO is largely missing. A clear explanation of the drawbacks of the previous DLR system is required. And the motivation to use the picosecond laser in the proposed work. A thorough performance comparison with the existing DLR facilities/methods is required.

Minor comments- 
Figure 5, 20- hardly gives any information. Probably use graphics/texts to make the figure more self-explanatory.

Figure 11, 12- whats the redline indicates here, please add an explanation in the caption.

Most of the figure's captions are also not very sufficient, provide detailed and self-explanatory captions.

Reviewer 2 Report

In this paper, the authors present the space DLR at the SHO.

Authors extensively revised the paper and added technical representations of their works. However, most of this paper describes the development history of their DLR like a survey paper. This paper is more suitable for a conference paper.